# A Dominant-Negative Mutant of ANXA7 Impairs Calcium Signaling and Enhances the Proliferation of Prostate Cancer Cells by Downregulating the IP3 Receptor and the PI3K/mTOR Pathway

**DOI:** 10.3390/ijms24108818

**Published:** 2023-05-16

**Authors:** Meera Srivastava, Alakesh Bera, Ofer Eidelman, Minh B. Tran, Catherine Jozwik, Mirta Glasman, Ximena Leighton, Hung Caohuy, Harvey B. Pollard

**Affiliations:** Department of Anatomy, Physiology and Genetics, Institute for Molecular Medicine, Uniformed Services University of Health Sciences (USUHS) School of Medicine, Bethesda, MD 20814, USA

**Keywords:** ANXA7, dominant-negative triple mutant (DNTM), IP3, mTOR, PI3K

## Abstract

Annexin A7/ANXA7 is a calcium-dependent membrane fusion protein with tumor suppressor gene (TSG) properties, which is located on chromosome 10q21 and is thought to function in the regulation of calcium homeostasis and tumorigenesis. However, whether the molecular mechanisms for tumor suppression are also involved in the calcium- and phospholipid-binding properties of ANXA7 remain to be elucidated. We hypothesized that the 4 C-terminal endonexin-fold repeats in ANXA7 (GX(X)GT), which are contained within each of the 4 annexin repeats with 70 amino acids, are responsible for both calcium- and GTP-dependent membrane fusion and the tumor suppressor function. Here, we identified a dominant-negative triple mutant (DNTM/DN-*ANXA7J*) that dramatically suppressed the ability of ANXA7 to fuse with artificial membranes while also inhibiting tumor cell proliferation and sensitizing cells to cell death. We also found that the [*DNTM*]ANA7 mutation altered the membrane fusion rate and the ability to bind calcium and phospholipids. In addition, in prostate cancer cells, our data revealed that variations in phosphatidylserine exposure, membrane permeabilization, and cellular apoptosis were associated with differential IP3 receptor expression and PI3K/AKT/mTOR modulation. In conclusion, we discovered a triple mutant of ANXA7, associated with calcium and phospholipid binding, which leads to the loss of several essential functions of ANXA7 pertinent to tumor protection and highlights the importance of the calcium signaling and membrane fusion functions of ANXA7 for preventing tumorigenesis.

## 1. Introduction

Annexin A7/ANXA7 is a tumor suppressor gene (TSG) that is located on chromosome 10q21 and mediates a variety of cellular functions, including membrane trafficking, signal transduction, and cytoskeletal organization [1,2]. Evidence for its tumor suppression function has been found in multiple types of tumors, including breast, ovarian, prostate, and pancreatic cancer, where it is frequently deleted or downregulated [3]. However, the mechanism through which ANXA7 functions as a tumor suppressor gene remains to be fully elucidated.

ANXA7′s tumor suppressor function was discovered serendipitously, when ANXA7 knockout mice were found to be laden with tumors [4]). Prior to that discovery, ANXA7 had been first isolated as a protein catalyst that mediated calcium- and GTP-dependent aggregation and the fusion of chromaffin granules, as well as the fusion of membranes and phospholipids [1,2,5,6]). The four C-terminal endonexin-fold GX(X)GT domains in ANXA7 have previously been determined to be responsible for calcium and GTP binding and subsequent membrane fusion functions [2]. ANXA7 has subsequently been detected in a variety of other mammalian tissues, including the brain, thyroid, liver, parotid, spleen, lung, and skeletal muscle [2] (Appendix A). Alternative splicing gives rise to two isoforms of 47 and 51 kDa. Most tissues harbor the lower-molecular-weight isoform. However, both isoforms are found in the brain and heart, and the larger isoform is exclusively expressed in mature skeletal muscle [7]. ANXA7 is detected in the cytosol, at the plasma membrane, around the nucleus, and at vesicular structures [8,9,10].

The loss of ANXA7 expression has been found to play a prognostic role in human cancers, including tumors from hormone-responsive tissues such as prostate and breast tissues [3,11,12,13,14,15,16]. Functionally, ANXA7 has been implicated in membrane trafficking, exocytosis, calcium homeostasis, and tumor suppression [3,11,15,17,18,19,20,21,22,23,24,25,26,27]. However, how ANXA7 promotes different cancers remains to be fully understood. Several recent studies have indicated that the loss of ANXA7 may promote the cell cycle, cell proliferation, and cell-adhesion-mediated drug resistance of multiple myeloma [25]. In addition, the loss of ANXA7 has been shown to promote epithelial–mesenchymal transition and to contribute to aggressiveness in hepatocellular carcinoma [28,29]. Finally, ANXA7 may have a bimodal role in the prognosis of different cancers [3,11,30,31,32,33]. However, how these properties are associated with the possible loss of calcium and the GTP dependence of membrane fusion is so far a fundamental mystery.

Therefore, we hypothesized that the 4 endonexin-fold motifs in ANXA7, which are contained within each of the 4 C-terminal 70-amino-acid-long annexin repeats, are responsible for both calcium- and GTP-dependent membrane fusion and the tumor suppressor function (see Figure 1). To test this hypothesis, we generated mutant *ANXA7* constructs with single or multiple mutations adjacent to all four of the endonexin-fold repeats [30]. The aim of these mutations was to change the charge associated with each of the endonexin motifs. We were then able to ask whether any combination of endonexin-fold mutations (i) suppressed the ability of ANXA7 to fuse artificial membranes; (ii) inhibited proliferation of prostate cancer cells; (iii) sensitized prostate cancer cells to cell death; and (iv) identified signaling pathways that were responsible for the tumor suppressor function.

The findings of this study are very important because we identified a dominant-negative triple mutant (DNTM) that yielded positive answers to all four questions above and concluded that at least three of the four C-terminal endonexin-fold **GX(X)GT** motifs were responsible for both calcium- and GTP-dependent fusion properties and the tumor suppressor gene functions. In particular, the results of this study indicate that variations in phosphatidylserine exposure, membrane permeabilization, and cellular apoptosis by wt-ANXA7 and DNTM variant were associated with differential IP3 receptor expression and PI3K/AKT/mTOR modulation in prostate cancer cells. Overall, this study provides valuable insights into the molecular mechanisms involved in the calcium- and phospholipid-binding properties of ANXA7 as a tumor suppressor and highlights the importance of the calcium signaling function of ANXA7 in preventing tumorigenesis.

## 2. Results

### 2.1. Amino Acid Residues Important for PS Liposome Aggregation

The endonexin-fold motif (GX(X)GT) in each repeat in the C-terminal domain is highly conserved among all annexins in many species (Figure 1). A similar electrostatic charge pattern associated with the GX(X)GT motif suggests that it could be responsible for the maintenance of the predominantly negative electrostatic potential on the C-terminal core around positively charged calcium ligands and negatively charged PS ligands. The replacement of polar residues (Asp or Glu → Ala) has been shown to inactivate phospholipid binding in ANXA4 [34]. ANXA4 is similar to ANXA7 in having an exceptional double-PS consensus site [35]. Thus, using charge-reversing mutagenesis, we expected to affect ANXA7 PS liposome aggregation properties in vitro and thereby reveal the specific roles of different repeats in the overall Ca and PS affinity of ANXA7. Using the DeepView/Swiss-PdbViewer, v3.7, we simulated the ANXA7 protein using ANXA1 crystallographic modeling. Electrostatic potentials were computed, and the modeled GX(X)GT motifs in wt-ANXA7 repeats showed a dipole-type distribution with a strong negative charge around the 3′-end (mostly Asp or Glu, underlined positions). As expected, the D^®^N and E^®^Q point substitutions of the 3′-end residues disrupted charge distribution in the modeled mutant motifs (Figure 2A,B). The disruption of the 3′-end polar residues (underlined positions) in annexin repeats can affect the Ca- and phospholipid-binding properties of ANXA7 by reversing electrostatic potential. This simulation of ANXA7 structural modeling with electrostatic potential distribution justified our choice of site-directed mutagenesis in order to evaluate the membrane aggregation properties by using a PS liposome aggregation assay under variable Ca concentrations and comparing the results with those found for the wt-ANXA7 and designed ANXA7 mutants.

To determine whether the substitution mutations changing the electrostatic potential distribution in wt-ANXA7 affected membrane fusion, we first mutated aspartic acid to asparagine (D to N) and glutamic acid to glutamine in each of the four repeats and introduced single and multiple mutations (Table 1). The mutants were expressed in E. coli and analyzed for artificial membrane fusion using liposomes. Subsequent analysis showed that mutations at single sites at repeat 1(Mnnn), repeat 2 (nMnn), and repeat 4 (nnnM) displayed properties that were similar to the properties of the wild-type protein (Figure 2C). However, mutations at repeat 3 (nnMn) resulted in the loss of ANXA7 activity. We reasoned that it was possible that a mutation at one site might affect neighboring sites in such a way as to affect the whole protein’s functionality. Therefore, we examined a number of double or triple mutants for their PS liposome aggregation activity. While mutations at repeats 1 and 2 (mutant F) or at repeats 3 and 4 (mutant K) rendered ANXA7 inactive, mutations at repeats 2 and 4 (mutant I) and mutations at repeats 1 and 4 (mutant L) left ANXA7 active. These results suggest that mutation at repeat 3 is important for membrane fusion. The functionality of mutants at repeats 2 and 3 (mutant C) as well as at repeats 1 and 3 (mutant H) could not be verified because the majority of proteins were bound to the cell membrane and could not be purified effectively. In order to further investigate the specific effect of the mutated repeat 3, we assessed the PS liposome aggregation mediated by triple and quadruple mutants (mutants G (repeats 1, 2, and 3); J (repeats 2, 3, and 4); O (repeats 1, 2, and 4); N (repeats 1, 3, and 4); and M (repeats 1, 2, 3, and 4), respectively) (Figure 2C). While the triple mutants such as J and N, containing repeat 3 mutation in different combinations with repeats 1, 2, and 4, had no PS liposome aggregation activity, mutant G with mutations at repeats 1, 2, and 3 had partial restoration of ANXA7 activity. However, the triple mutant O, with no site mutated at repeat 3, and mutant M with all sites mutated displayed properties that were similar to the properties of the wild-type protein. The choice of site-directed mutagenesis thus revealed a functional diversity of repeats in ANXA7.

ANXA7 monomers combine to form dimers in order to facilitate membrane fusion. Therefore, we assessed the dominant-negative effect of the mutants with a repeat 3 mutation, and thus with negligible PS liposome aggregation activity (mutants J and H). Only mutant J abolished the wt-ANXA7-mediated aggregation, as demonstrated in assays containing the wt-ANXA7 in a mixture with mutant J (Figure 2C, insert). These results suggest that the mutated monomer in mutant J blocks the dimerization process through unique positioning and interaction with the N-terminal domain, which may be responsible for the marked decrease in liposome aggregation. The relationship between the mutated site, protein conformation, and the membrane fusion gene function is illustrated in Figure 3.

### 2.2. Dominant-Negative ANXA7J Does Not Kill Prostate Cancer Cells

Our studies with both human cancer cells and the *Anxa7* knockout mouse have indicated the existence of a possible common deficit in calcium regulation [32]. Furthermore, we have also shown that the altered copy number for *ANXA7* is associated with worse prostate prognosis (Appendix A). In the case of cancer cells, it is known that the initial signal for the onset of programmed cell death is the release of a pulse of calcium from the internal calcium stores in the endoplasmic reticulum. Therefore, we tested prostate cancer cells for the effect of the wild-type and the dominant-negative triple mutant *ANXA7* on the cytotoxicity assay. Adenovirus vectors containing either the wild-type *ANXA7* (wt-*ANXA7)* gene or the dominant-negative mutant *ANXA7J* (DN-*ANXA7J*) were transfected into the DU145 prostate cancer cell line and tested for their growth inhibition. Initially, the cells were infected with 10, 20, or 40 pfu/cell of either control adenovirus or adenovirus expressing wt-ANXA7 or DN-ANXA7J (Figure 4). The cells were analyzed for their growth and counted using a hemocytometer. The experiments were carried out in triplicate for 24 h. The cytotoxicity assay revealed that in DU145 cells, at 24 h, the vector alone and DN-ANXA7J had similar effects, while the addition of wt-ANXA7 killed the cancer cells effectively (Figure 4). Since the DN-ANXA7J is against the calcium binding site and the addition of this mutant killed the cancer cells, we conclude that calcium associated function with this identified mutation may be mechanistically involved in the tumor suppression phenotype.

### 2.3. Impact of DN-ANXA7J on Apoptosis, Cell Cycle, and Cell Morphology

Since ANXA7 can potently suppress the proliferation of DU145 prostate cancer cells, we extended the study to test whether transiently transfected wt-*ANXA7* promotes the apoptotic state of DU145 cancer cells. We used the TUNEL assay and compared normal prostate epithelial cells (PrECs), as well as cells transfected with DN-*ANXA7J*. The ratios of apoptotic cells were determined by detecting fragmented DNA. The cells were stained using an APO-BRDU Kit (BD Pharmingen, San Diego, CA, USA). At 24 h after the infection, both attached and floating cells (1 × 10^6^) were harvested and processed according to the manufacturer’s protocol, with the exception that we substituted the PE-conjugated mouse anti-BrdU monoclonal antibody (BD Pharmingen, San Diego, CA, USA) for the kit’s FITC-labeled antibody in order to analyze GFP fluorescence and anti-BrdU PE labeling simultaneously. The cells were analyzed using an EPICS XL-MCL flow cytometer (Figure 5A). The number of apoptotic DU145 prostate cancer cells increased with the addition of wt-ANXA7 from 5% to 39% with DN-ANXA7J. The increase in apoptosis with the addition of p53 was 48%. No cell death was observed in PrEC cells with the addition of wt-ANXA7, although a 10% apoptosis was observed in the presence of p53.

The diverse effects of *ANXA7*, DN-*ANXA7J*, and p53 on cell death/proliferation implied their distinct involvement in the regulation of the cell cycling program. Wt-ANXA7 caused a similarly prolonged G2 phase (up to ~15%) in hormone-resistant PrCa (Figure 5B). Consistent with the shielding effect of wt-ANXA7 on benign cell death, the wt-ANXA7-dependent G2 arrest was substantially diminished in PrEC compared with cancer cells. Remarkably, DN-*ANXA7J,* which failed to induce PCD, did not cause essential changes in cell cycling. This result suggests that calcium signaling regulated by ANXA7 has a major role to play during the cell cycle.

Recent studies demonstrated that the calcium released from the endoplasmic reticulum synchronizes the mass exodus of cytochrome *c* from the mitochondria, a phenomenon that coordinates apoptosis. Thus, we decided to determine whether these properties were altered in DN-*ANXA7J*-transfected cells. In studies with the DU145 cancer cell line, we found that the effect of wt-ANXA7, but not DN-ANXA7J, on the enhanced release of cytochrome c had a threefold increase in comparison to the vector-alone control. By contrast, the use of p53 as a positive control revealed only a twofold increase in the release of cytochrome c (Figure 6A). These results suggest that the ANXA7-induced apoptotic pathway involves cytochrome c release, indicating the likely involvement of the mitochondria. In addition, the cells transfected with wt-*ANXA7* were morphologically different with GFP fluorescence on the membranes. On the other hand, GFP fluorescence was distributed throughout the vector-alone control or DN-*ANXA7J* transfection (Figure 6B). Wt-ANXA7 induced morphological changes, including cell shrinkage.

### 2.4. Effect of DN-ANXA7J on IP3 Receptors

Studies of Ca^2+^ metabolism in beta cells from *Anxa7* (+/−) knockout mice have shown that thapsigargin failed to raise cytosolic Ca^2+^ and failed to activate SOC channels. The IP3 ligand also failed to release intracellular Ca^2+^ from the ER. We have reported that the reason for the lack of efficacy of thapsigargin in the *Anxa7* (+/−) knockout mouse is a documented tenfold deficiency in IP3 receptors [32]. The importance of IP3 receptors in cancer cells is that the activation of IP3 receptors by IP3 is the physiological stimulus needed to release calcium from the ER, thus triggering the mitochondrial permeability transition, which leads to apoptosis [36]. Therefore, we hypothesized that the action of the transfected *ANXA7* gene on tumor cells may be to elevate cytosolic Ca^2+^ and to potentiate subsequent proapoptotic actions of the released calcium. We conjectured that DN-ANXA7J might have the opposite effect. We, therefore, measured the mRNA levels of *ANXA7* and IP3 receptor in tumor cells treated with the adenovirus vector alone, as well as with wt-*ANXA7* and DN-*ANXA7J*. Quantitative RT-PCR was used to quantitate the IP3 receptor messages (types 1, 2, and 3), and the levels of the beta-actin message were used to normalize the levels of RNA used for the analyses. As is shown in Figure 7, DN-*ANXA7J* downregulated all three IP3 receptor expressions in the DU145 metastatic prostate cancer cell line. We found that wt-*ANXA7* expression increased Ad vector-alone control from 0.49 to 0.70, and we observed a corresponding increase in IP3 receptor expression. By contrast, the dominant-negative mutant *ANXA7J* expression suppressed IP3 receptor expression by 0.80, since it inhibited the activity of endogenous *ANXA7*. These results suggest that *ANXA7* controls all three subtypes of IP3 receptor expression and thus calcium signaling.

### 2.5. Identification of Downstream Targets of ANXA7 That Constitute the Calcium Signaling Pathway Involved in Tumorigenesis

To determine the association of ANXA7 with other proteins in the cellular system, we used GO annotation. The RNA data were used to cluster genes according to their expression across different cellular signaling clusters (Appendix A). In order to assess the downstream targets of wt-*ANXA7*, DN-*ANXA7J*, or p53 in prostate cancer cells, we isolated mRNAs from the normal prostate epithelial cell line PREC and from the prostate cancer cell DU145, which had been transfected with either the vector alone, wt-*ANXA7*, DN-*ANXA7J*, or p53. The tumor-specific gene expression profiles in DU145 cells transfected with wt-*ANXA7*, DN-*ANXA7J*, or p53 were determined using cDNA microarray analysis (Atlas Human Cancer 1.2 Arrays and AtlasImage 2.01 software, Clontech, Palo Alto, CA, USA). Adjusted intensities were calculated as spot intensities (minus background values for the spots) multiplied by global normalization coefficients.

Table 2 provides the list of the genes in metastatic DU145 cells whose expression levels were most corrected towards the respective expression levels in the normal PREC prostate cell line due to the effect of DN-*ANXA7J,* wt-*ANXA7*, and P53. The genes are ordered according to our list of priorities for validating their involvement in the wt-*ANXA7*-induced effects on prostate cancer progression. These data are graphically shown in Figure 8, which shows the expression levels in DU145 cells (as the log ratio to the expression in the normal PREC prostate cell line) after treatment with DN-*ANXA7J*, wt-*ANXA7*, P53, or the vector alone. The x-axis shows the average expression level of the respective genes in the metastatic DU145 and PREC cells. The transfected DU145 cells are depicted with magenta diamonds (for DN-*ANXA7J*), blue diamonds (for wt-*ANXA7*), and green diamonds (for P53). The improvement in the expression levels is reflected in the movement of the expression in the transfected cells towards the x-axis. The movement of a gene towards the x-axis means that it tends to show equivalent relative expression in PREC normal cells and transfected metastatic DU145 cancer cells with DN-*ANXA7J*, wt-*ANXA7*, or P53. These data thus indicate that the wt-*ANXA7* or p53 treatment of metastatic DU145 cells leads to their significant resemblance to PREC normal cells. No significant difference was observed between DN-*ANXA7J* transfection in correcting the aberrant gene expression in cancer cells. This means that we need to use other mutants against GTP and PKC or a combination of these in order to differentiate the requirements of ANXA7 action. On top of the lists are the genes related to apoptosis and tumor suppression (S6KII-alpha 3 and BAK), cell cycle (CDC25C), and metastasis and invasiveness (Integrins, TGF-beta2, c-myc, and MMP-9).

Considering the possibility of ANXA7 involvement in arachidonate-regulated phospholipid aggregation and exocytosis, we explored the effects of wt-*ANXA7* or DN-*ANXA7* on the lipid-relevant gene expression in human prostate cancer cells. Wt-*ANXA7*-induced changes were characterized as differences in the adjusted intensities from direct comparisons to DN-*ANXA7J* in the corresponding arrays. As expected, wt-*ANXA7* demonstrated distinct effects on the lipid-relevant gene expression network in cancer cells. Wt-*ANXA7* downregulated PI3 kinase p85-alpha, *PI3K type 3 catalytic subunit p100 (PIK3C3),* phosphatidylcholine-hydrolyzing *phospholipase PLD1,* PI3K-related Ser-Thr protein kinase *FRP1*, and *DAG kinase-epsilon* (Figure 9A). PIK3C3 is responsible for phosphatidylinositol-3-phosphate synthesis and is essential for phagolysosome formation. Directly implicated in autophagy regulation and crucial for cell growth control, it was found to regulate intracellular vesicle trafficking and mediate mTOR activation in response to amino acid availability, as reviewed in [37].

To detect the possible mTOR involvement in the tumor suppressor effects of ANXA7 in prostate cancer cells, we juxtaposed the microarray-derived mTOR gene expression and apoptotic rates based on PS exposure and cell membrane permeabilization (ANXAV-PE, Figure 9B). Consistent with the earlier-demonstrated shielding effect on benign cells, wt-ANXA7 exclusively eliminated cancer cells by inducing apoptosis, which was accompanied by PS exposure on the external membrane leaflet of the dying cells. Most remarkably, this differential effect of wt-ANXA7 was associated with moderate mTOR inhibition. By contrast, the dominant-mutant DN-*ANXA7J* (which failed to induce PS liposome aggregation or enhance PCD) caused an equally low PS exposure (only ~50% of the wt-ANXA7 levels) in benign and cancerous cells. The PCD resistance of cancer cells to DN-*ANXA7J* was paralleled by mTOR upregulation, whereas similar apoptotic rates in response to p53 in both benign and cancer cell types were accompanied by equally reduced mTOR levels.

Thus, the aggregation and fusion properties of the ANXA7 PS membrane can be directly linked to the ANXA7-mediated cancer-specific cell elimination. Moreover, ANXA7-attributed PS exposure in PCD can be specifically implicated in mTOR-controlled cell survival. Further studies may elucidate pleiotropic ANXA7 effects on cell death/proliferation that are likely to share autophagy-relevant mechanisms with the major PI3K/Akt/mTOR players detected in cancer.

## 3. Discussion

In this study, we tested the hypothesis that the 4 endonexin-fold motifs in ANXA7, which are contained within each of the 4 C-terminal 70-amino-acid-long annexin repeats, are responsible for both calcium- and GTP-dependent membrane fusion and the tumor suppressor function. Annexins have been implicated in cell–cell fusion [38,39]. By investigating all the possible combinations, we determined that reversing the charged residues at the 3′ ends of the endonexin-fold motifs in three of the four annexin repeats resulted in the loss of membrane aggregation and fusion properties. Furthermore, the relevance of ANXA7 to prostate cancer was also emphasized in this study. From the results of our previous studies and the current study, we found that the loss of ANXA7 is associated with the worst forms of prostate cancer [11,17,40]. Indeed, ANXA7 is a tumor suppressor gene at position 10q21, and the copy number alteration (CNA) is associated with a worse prostate cancer prognosis (Appendix A). However, there is no clear understanding of how the loss of ANXA7 induces different stages of cancer progression. Our current findings suggest that the wt-ANXA7 was able to inhibit the proliferation of prostate cancer cells; enhance the sensitized prostate cancer cells, leading to cell death; and identify signaling pathways that were responsible for the tumor suppressor function. The results suggest that the glutamic and aspartic acid residues involved in calcium and phospholipid binding serve as the determinants of ANXA7 action on IP3 receptor expression and prostate cancer cell survival via the PI3K/mTOR pathway. Overall, this study provides support for the likelihood that there is a close association between the loss of calcium-dependent membrane fusion and enhanced prostate cancer cell growth.

### 3.1. Endonexin-Fold Motif Mutations and Inhibition of Aggregation and Membrane Fusion

In this paper, we identified negatively charged 3′-end residues in a novel GX(X)GT motif shared by all ANXA7 repeats that can control electrostatic potential distribution as a major tool in orchestrating the N-terminal positioning and binding of Ca and PS ligands to the ANXA7 core. The highest overall affinity for the PS-binding consensus site in repeat 2 as well as a unique role of repeat 3 in Ca dependency and PS binding were observed through ANXA7 protein simulation and confirmed via the PS liposome aggregation assay in vitro. As repeat 3 harbors a putative “capping” E398 and inserted N-terminal, it can act as a switch in ANXA7 structural rearrangement and has a dominant regulatory effect in the ANXA7-mediated membrane aggregation.

In this study, we found that the ANXA7 repeats control the electrostatic structure of the protein. Why is this important? A possible reason is that the charge on the plasma membrane is “−” on the outside and “+” on the inside. By switching from the net “−” to the net “+” when mutated, ANXA7 cannot bind to the inner layer of the plasma membrane and thus fails to start fusion and initiate the formation of cell–cell fusion, which is critical for cancer.

The structural template for the current ANXA7 simulation is derived from ANXA1, the most scrutinized member of the annexin family, which is specifically known for the diverse effects of its N-terminal tail [41]. Since ANXA1 was recently found to be downregulated in multiple human cancers in vivo [42,43,44,45], the fact that the ANXA1 N-terminal associates with cancer biomarkers cytokeratins 8 and 18 [46] could provide some insights into tumor suppression by other annexins, including ANXA7. As suggested by ANXA7 protein simulation, a dominant-negative effect of the DN-ANXA7J construct on PS liposome aggregation may result from the loss of N-terminal structural flexibility, which has an impact on ANXA7-associated downstream effects. Since *ANXA7* can act as a Ca-activated GTPase affecting exocytosis and phospholipid membrane fusion [18,19,20,21,22,23,24,25,26], its annexin family member properties (which could include phosphatidylserine exposure on dying cells) appeared to contribute to ANXA7 control over DU145 cell survival. DN-ANXA7J, which can inhibit wt-ANXA7-mediated phosphatidylserine liposome aggregation, lacked tumor suppressor effects. These data implicate a role for ANXA7 Ca-/phospholipid-binding properties on prostate tumorigenesis. Using DU145 as a prostate cancer (PrCa) model where wt-ANXA7 and DN-ANXA7J displayed the most contrasting effects, we found that the annexin properties of wt-ANXA7/DN-ANXA7J can be further implicated in the cell survival modulation. Consistently, in this study, wt-ANXA7 and DN-ANXA7J distinctly interfered with apoptosis as well as arachidonate lipoxygenation in prostate cancer cells [16]. Indicating possible PCD-associated mechanisms, DN-ANXA7J caused the loss of PS liposome aggregation, corresponding to the failed downregulation of 15-lipoxygenases in response to corticosteroids or IL-4. Thus, the preferential PS affinity of ANXA7 and its ability to aggregate PS liposome membranes implicated ANXA7 in the facilitation of the membrane phospholipid asymmetry and the PS-relevant PCD in particular, which could constitute major tumor suppressor effects of ANXA7 on cell death and proliferation.

### 3.2. Inhibition of Proliferation of Prostate Cancer Cells

Defects in intracellular Ca2+ regulation may in fact underlie phenotypic manifestations in ANXA7 (+/−) mice, ranging from neoplasia to anxiety. Our studies with prostate cancer cells imply that the overexpression of ANXA7 increases the IP3 receptor expression, and the dominant-negative ANXA7J mutant downregulates the IP3 receptor expression. Reduced *Anxa7* gene dosage in the *Anxa7* heterozygous mice not only leads to an aberrant Ca^2+^ effect on insulin secretion but also results in a selective decrease in IP3 receptor expression and IP3-receptor-mediated calcium stores, as well as a defective store-operated calcium channel (SOC) [32]. Recent studies have identified Ca^2+^-dependent mechanisms involved in the control of cell growth and apoptosis of human prostate cancer cells and have characterized SOCs in these cells [47,48]. For years, it was believed that Ca^2+^-related apoptosis may be triggered by large, sustained increases in cytosolic Ca^2+^ [49]. However, other studies indicate that the Ca^2+^ status of the ER lumen also affects cell sensitivity to apoptosis [50,51,52,53] and that the overexpression of Bcl-2 or the underexpression of ANXA7 triggers a reduction in the ER Ca^2+^ content due to an increased leak [54,55,56,57]. The fact that Bcl-2 and ANXA7 modulate the Ca^2+^-filling status of the ER strongly suggests that the oncoprotein (Bcl-2) and the tumor suppressor (ANXA7) may interfere with the capacitative Ca^2+^ influx as well, which is activated through ER depletion [56]. Capacitative Ca^2+^ entry is mediated by store-operated channels (SOCs) generating Ca^2+^ current, _I_SOC. SOCs, located in the plasmalemma, may be activated by a variety of active or passive mechanisms, all of which share the property of depleting ER Ca^2+^ stores. Ca^2+^ entry via SOCs induces a sustained increase in cytosolic Ca^2+^ concentration ([Ca^2+^]c), which helps to replenish ER. Therefore, when activated, SOCs regulate both cytoplasmic and ER intraluminal ([Ca^2+^] ER) free Ca^2+^ concentrations. For this reason, SOCs are increasingly attracting attention as potential apoptosis regulators [58,59].

### 3.3. Sensitization of Prostate Cancer Cells to Cell Death

In this study, we described the role of ANXA7 in cellular fusion and how it is relevant to the trafficking and eventually the link with prostate cancer. Membrane fusion is a process that involves the merging of two cellular membranes to form a single continuous membrane. This process is critical for various cellular processes, including vesicle trafficking, neurotransmission, and fertilization. The dysregulation of membrane fusion has been linked to various diseases, including cancer. In cancer cells, abnormal membrane fusion events can lead to uncontrolled cell growth, invasion, and metastasis. Moreover, several proteins involved in membrane fusion have been implicated in cancer, including SNARE proteins, annexins, and fusogenic viral proteins.

The modulation of ANXA7 structural flexibility, including N-terminal positioning, is a promising approach for further evaluation of ANXA7 in PS-associated PCD and tumor suppression. In comparison with ANXA1, ANXA7 can serve as a hormone-responsive endogenous modulator of phospholipid-relevant cell survival and eicosanoid generation. Recent studies have linked ANXA7 to hormone transport and prostate cancer. ANXA7 is known to interact with the androgen receptor (AR), a key regulator of prostate cancer growth. It is important to mention that the expression of ANXA7 is much higher in endocrine tissues such as the thyroid and parathyroid glands (Appendix A). In normal prostate cells, ANXA7 is thought to modulate AR activity by inhibiting its translocation to the nucleus. However, in prostate cancer cells, ANXA7 is often downregulated, leading to increased AR signaling and cancer cell proliferation [60]. ANXA7 has also been shown to play a role in the transport of estradiol, a key estrogen hormone, in breast cancer cells. Similarly, in prostate cancer, ANXA7 may play a role in the transport of androgens, which are essential for prostate cancer growth [60]. Similar to in vivo regulatory effects of ANXA1 peptides, ANXA7 protein segments could have therapeutic potential in cancer or other pathologies that involve PS membrane dynamics with phospholipid-associated signaling and metabolism.

Annexins have common implications in exocytosis, apoptosis, and arachidonate cascade. In the tightly regulated process of cell division, major checkpoints (G1-S and G2-M) delay progression into the next cell cycle phase until the previous step is fully completed. Therapeutic approaches in cancer employ the modulation of cell cycling and the induction of apoptosis or cell growth arrest by many antitumor drugs that are cell-cycle-specific. Major differences in cell cycling (the G2 arrest under wt-ANXA7 versus the G1 arrest and the S-phase delay under p53 and no essential changes with DN-ANXA7J), echoed the distinct character of the wt-ANXA7 effects on cell death/proliferation compared with its dominant-negative form (DN-ANXA7J) or the conventional cell-cycle regulator p53. While the end of and G1 phase is characterized by the highest sensitivity to cytotoxic effects, the G2 phase includes immune surveillance and DNA damage repair, which could be involved in the predominant G2 arrest by wt-ANXA7. As an inhibitor of the G2 checkpoint that can selectively sensitize cancer cells with inactive p53 to DNA-damaging drugs or ionizing radiation, ANXA7 may have specific therapeutic potential in many human cancers.

### 3.4. Identification of Signaling Pathways That May Be Responsible for the Tumor Suppressor Function

Considering ANXA7 involvement in the arachidonate-regulated phospholipid aggregation and exocytosis, as well as its suggested regulation by steroids, we explored the effects of wt-*ANXA7* or mutant-*ANXA7* on the lipid-relevant gene expression in human prostate cells, specifically representing the in vitro models of the steroid sex-hormone-sensitive tumorigenesis. We used the cDNA microarray platform to obtain the gene expression profile. Wt-*ANXA7*-induced changes were characterized as differences in the adjusted intensities from direct comparisons with DN-*ANXA7J* in the corresponding arrays. In addition, the phospholipid-binding ANXA7 had distinctive effects compared with the canonical cell-cycle regulator p53. As expected, wt-*ANXA7* demonstrated distinct effects on the lipid-relevant gene expression network in cancer cells. Most remarkably, wt-*ANXA7* induced tissue-specific alterations in the autophagy-relevant insulin/PI3K/Akt/mTOR signaling cascade (including also *FOXO* and *PTEN*) that has a central role in carcinogenesis as one of the most mutated systems in human cancer [37].

*PI3K* changes were present in cancer arrays: wt-*ANXA7* inhibited the *PI3K type 3 catalytic subunit p100 (PIK3C3*) (Figure 9A). The PI3K/Akt/mTOR network (reviewed in [61] and [62]) includes PI turnover, steroid control, and insulin signaling, all of which could be associated with ANXA7. PIK3C3 is responsible for phosphatidylinositol-3-phosphate synthesis and is essential for phagolysosome formation. Directly implicated in autophagy regulation and crucial for cell growth control, it was found to regulate intracellular vesicle trafficking and mediate mTOR activation in response to amino acid availability [37].

Interestingly, in our *Anxa7* (+/−) murine model [14], defective insulin secretion was associated with a profoundly reduced protein expression of the IP3 receptor for PI-1,4,5-trisphosphate. Among three classes of widely expressed PI3Ks that show distinct substrate preferences in vivo, class I (including p85-alpha) primarily generates PI-3,4,5-triphosphate from its 4,5-biphosphate form, which can be hydrolyzed into PI-1,4,5-triphosphate and diacylglycerol. In the PI3K heterodimer, the regulatory p85-alpha that is crucial in mediating the activation of class IA PI3Ks through the growth factor receptor tyrosine kinases recruits the p110 catalytic subunit to the membrane, where p110 phosphorylates inositols. Responding to sex hormones, estrogen, and androgen, p85 can rapidly assemble a membrane-associated molecular complex with c-Src and the ER or AR [63,64]. Specifically involved in insulin signaling, p85-alpha is necessary for the insulin-stimulated increase in glucose uptake and glycogen synthesis, whereas rapamycin can abolish its mRNA upregulation by insulin [65].

Wt-*ANXA7* also downregulated phosphatidylcholine-hydrolyzing *phospholipase PLD1* as well as *diacylglycerol DAG kinase epsilon* in prostate cancer cells (Figure 9A). *DAG kinase epsilon* catalyzes diacylglycerol in the PI turnover and is also associated with arachidonic acid, which can promote ANXA7 membrane aggregation properties [23]. Highly selective for arachidonate-containing diacylglycerol species, DAG kinase epsilon can either terminate signaling through arachidonoyl-DAG or contribute to the synthesis of phospholipids with defined fatty acid composition [66]. By controlling the amount of its substrate DAG and the route it takes, DAG kinase epsilon could bridge the arachidonate cascade to the regulation of PI turnover and the PI3K pathway in response to wt-*ANXA7*.

Downregulated by wt-*ANXA7* in PrCa, *phospholipase PLD1* is also connected to the arachidonate cascade and PI turnover. Ubiquitously expressed in mammalian cells, PLD1 and PLD2 isoenzymes have been implicated in intracellular signal transduction (including Ca response), vesicle transport, endo/exocytosis, cell shape and migration, mitosis, and cytoskeletal reorganization. PLD1 is specifically localized to secretory granules, while PLD2 is localized to the plasma membrane. PLD can have antiapoptotic properties suppressing ceramide formation (reviewed in [67]), whereas PLD activity and PLD1 mRNA levels can be downregulated during apoptosis [68]. Consistently, PLD activity was found elevated in many human cancers, and transformed cells showed PLD substrate specificity for the phosphatidylcholine-lacking arachidonate acyl groups [69]. Moreover, the metabolic product of PLD (phosphatidic acid) has been recently implicated in mTOR regulation (reviewed in [70]). Therefore, *PLD1* downregulation by wt-ANXA7 could mediate its cell elimination effects (especially in prostate cancer). It is intriguing to suggest that the effects of wt-ANXA7 versus those of the dominant-negative DN-ANXA7J on the endogenous prostate-specific tumor suppressor 15-LOX2 in prostate cancer cells [16] could have incorporated PLD-mediated polyisoprenyl–phosphate signaling, which is a switch regulated by lipoxins and leukotrienes, the arachidonate autocoids with opposing responses in immunity and apoptosis [71].

Thus, the dissimilar PS membrane aggregation properties of wt-*ANXA7* and DN-*ANXA7J* were accompanied by a distinctive lipid-relevant syn expression that could be implicated in cell death and proliferation. Specific alterations in phospholipid turnover and arachidonate cascade under wt-*ANXA7* could involve the autophagy-relevant control over cell survival with different end results in prostate cancer cells compared with DN-ANXA7J.

Most remarkably, wt-*ANXA7* specifically induced apoptosis in cancerous (but not benign) cells, which was accompanied by PS exposure on their cell membrane and the inhibition of *mTOR* gene expression. The abortive PS exposure caused by DN-*ANXA7J* was concordant with the increase in mTOR in cancer cells. The analysis of cell death/proliferation and cell cycling with the corresponding cell-size alterations also indicated an essential role of both types (I and II) of PCD for wt-*ANXA7* but not the DN-*ANXA7J* mutant, which lacked the PS liposome aggregation properties. The Wt-*ANXA7*-associated lipid-relevant gene expression network substantiated its ties to phospholipid turnover, the arachidonate cascade, and the PI3K pathway in particular.

### 3.5. Calcium, *IP3 Receptor*, and Cancer

Calcium signaling is a crucial process in many cellular functions, including cell growth, proliferation, and apoptosis. The inositol 1,4,5-trisphosphate receptor (IP3R) is a calcium channel that is activated by the second messenger IP3, which is produced in response to extracellular signals. IP3R plays a critical role in calcium release from intracellular stores, which is important for various cellular processes. There is growing evidence to suggest that the dysregulation of calcium signaling, including IP3R, is involved in the development and progression of various types of cancer. For example, studies have shown that the increased expression of IP3R is associated with tumor growth and poor prognosis in breast and prostate cancer.

Large molecules such as proteins have been shown to have similar effects to those induced by thapsigargin. For example, exogenously added TGFβ also arrests cancer cells in G1/G0 and induces apoptosis [72]. The transfection of the *ANXA7* gene into cancer cells also has a very similar sequence of actions [73]. This fact, plus the high prevalence of tumors in the *Anxa7* (+/−) knockout mouse model and the disorders of calcium metabolism in *Anxa7* (+/−) mouse tissues appear to implicate a specifically thapsigargin-like mechanism for how the ANXA7 gene activates human tumor cell apoptosis. For example, in cancer cells, thapsigargin raises the cytosolic calcium concentration by preventing Ca^2+^ re-entry into the ER and activating SOC channels. Similar studies of Ca^2+^ metabolism in beta cells from *Anxa7* (+/−) knockout mice showed that thapsigargin failed to raise cytosolic Ca^2+^ and failed to activate SOC channels. The IP3 ligand also failed to release intracellular Ca^2+^ from the ER. So, the reason for the lack of efficacy of thapsigargin in the *Anxa7* (+/−) knockout mouse model is a documented tenfold deficiency in IP3 receptors [32]. The importance of IP3 receptors in cancer cells is that IP3 receptor activation by IP3 is the physiological stimulus needed to release calcium from the ER, thus triggering the mitochondrial permeability transition, which leads to apoptosis [36]. Localized in both cytosol and nucleus [74] and involved in Ca release through a major phospholipid (inositol-1,4,5-triphosphate, IP3) cascade [15], ANXA7 may regulate apoptosis by transporting PS to the plasma membrane leaflet versus the shuttling PS between the nucleus and cytosol.

*Anxa7* deficiency in mice also caused multiple endocrine defects, such as adrenal medullary hypertrophy and chromaffin cell hyperplasia. Langerhans islets in these mice exhibited defective intracellular Ca signaling, and insulin secretion was associated with the profoundly reduced protein expression of the IP3 receptor for one of the phosphatidylinositols (PI-1,4,5-trisphosphate). In addition to being a second messenger that mediates the release of intracellular Ca and thereby regulates various Ca-dependent processes, phosphatidylinositol (PI) is a part of the major PI3K/Akt/mTOR cascade regulating cell survival. Most importantly, using the gene expression profiling of Langerhans islets, we also found that *Anxa7* deficiency was associated with the inhibition of *PTEN* (a lipid phosphatase that downregulates the phosphatidylinositol 3-kinase (PI3K)) as well as additional changes in lipolytic and ubiquitin–proteasomal pathways [13,75]. The murine model thus indicated that tumor suppression by ANXA7, which is involved in the Ca-regulated phospholipid turnover, can be particularly associated with the PI3K pathway [76].

The specific regulatory control of the calcium signaling pathway occurs through the concomitant activation of other signaling pathways. Numerous isolated observations suggest that the PI3K/Akt pathway may also influence the calcium signaling pathway. Given that mTOR controls cell proliferation and cell homeostasis, and that calcium plays a key role in these two phenomena, it follows that mTOR facilitates IP3R-mediated calcium release when the nutritional status of cells requires it. Our results demonstrate a novel mTOR/PI3K signaling pathway where ANXA7 stimulates cytosolic calcium signaling through the regulation of IP3 receptor expression. This ANXA7/mTOR/ PI3K/IP3R signaling cascade elevates the cytosolic calcium concentration by releasing calcium from the internal ER stores. The cytosolic calcium response mediated through charged asp and glu residues in ANXA7 subsequently regulates IP3R expression and function. Our data also show that the mutation of these residues results in the reduced expression of mTOR and PI3K, resulting in impaired membrane fusion and cell survival. These data enhance the understanding of the multifactorial ANXA7 in regulating multiple signaling pathways and biological processes.

The IP3 receptor mediates calcium release from the ER [77]. An increase in cytosolic calcium can lead to apoptosis [78]. We, therefore, investigated if wt-ANXA7 increased the expression of the IP3 receptor. In the DU145 prostate cancer cell line, IP3 receptor expression was significantly elevated after exposure to wt-ANXA7 and decreased with DN-ANXA7J. An elevated IP3 receptor leads to an increase in the mitochondrial calcium concentration, which in turn opens the mitochondrial permeability transition pore (MPTP). This is followed by an efflux of cytochrome c, activating the intrinsic pathway of apoptosis [79]. Calcium signaling and IP3R can affect cancer development in multiple ways. One way is by promoting cell proliferation and survival, which are the key characteristics of cancer cells. Calcium signaling can also regulate cell migration and invasion, which are important for cancer metastasis. In addition, the dysregulation of calcium signaling can lead to genomic instability and DNA damage, which are hallmarks of cancer.

In our study, we found that wt-ANXA7 increased cytosolic cytochrome c, and DN-ANXA7J had almost no effect on Ca2+ homeostasis. These data are compatible with our previous findings demonstrating that the loss of Anxa7 in cells leads to the downregulation of apoptosis activation through the downregulation of the IP3 receptor, thereby leading to altered intracellular calcium homeostasis in Anxa7 knockout mice [32]. Therefore, targeting calcium signaling and IP3R may hold promise as a potential therapeutic strategy for prostate cancer.

## 4. Materials and Methods

### 4.1. Site-Directed Mutagenesis

Plasmids for wild-type *ANXA7* as well as multiple mutants were constructed using the pTrc99 vector system. Aspartic-acid-to-asparagine (D-to-N) and glutamic-acid-to-glutamine (E-to-Q) mutations in pTrc-FLS were generated using an overlap PCR method, with the pTrc99 plasmid [80] as the template. The primer sequences are available upon request. Briefly, PCR products were purified and digested with unique restriction enzymes (Sac1 and SnaB1 or SnaB1 and EcoRI) and used for subcloning into plasmid pTrc-FLS [80]. Site-directed mutations were introduced into the calcium-binding sites as combinations of all four crystallographic-defined endonexin-fold motifs. All four motifs have a consensus sequence (GXGTDE), and the amino acid residue mutations were engineered to generate the amidated analogs of the underlined charged residues (viz., (GXGTNQ)). We prepared 16 different combinations, including the wild-type ANXA7. The combinations were single mutations (e.g., 1, 2, 3, or 4 repeats); mutations at two sites (e.g., 1 and 2, 1 and 3, etc.); mutations at three sites (e.g., 1 and 2 and 3, 2 and 3 and 4, etc.); and all four sites (e.g., 1 and 2 and 3 and 4). Specific mutations that were incorporated via polymerase chain reaction using Pfu Turbo DNA Polymerase (Stratagene) were confirmed using DNA sequencing.

### 4.2. Preparation and Purification of ANXA7 Proteins In Vitro

The human recombinant wild-type and mutant ANXA7 were isolated and purified as described previously [17,80]. Briefly, *Escherichia coli* bacteria containing the ANXA7-expressing vector were grown in 1 L of Luria broth at 37 °C. After reaching an *A*540 level of 0.6, the culture was incubated overnight in the presence of 1 mM isopropyl-b-D-thiogalactopyranoside (IPTG). After incubation, the bacteria were harvested via centrifugation (1000× *g* for 10 min; Beckman Coulter Allegra X-15R). The expressed recombinant ANXA7 was then extracted from the *E. coli* paste, concentrated via precipitation with 0–20% (*w*/*v*) (NH4)_2_SO_4_, and purified via gel filtration using Ultragel AcA54 (Biosphere). The partially purified ANXA7 was further purified by binding to PS lipid vesicles in the presence of Ca^2+^ and extracting with EGTA. This purification step was repeated six times, followed by column chromatography on Ultragel AcA54 to finally yield a highly purified (~99%) ANXA7, determined via SDS–PAGE and silver staining.

### 4.3. Preparation of Phosphatidylserine-Containing Liposomes for Aggregation Assay In Vitro

Phospholipids (PS) were obtained from Avanti Polar Lipids. PS lipid vesicles were prepared fresh daily using the swelling method [81]. Briefly, highly purified (0.99%) brain phosphatidylserine (Avanti Polar Lipids) in a 1:4 chloroform–methanol solution was slowly dried under nitrogen and then allowed to swell in 0.3 M sucrose at room temperature. The suspension was then sonicated and centrifuged at 12,000× *g*. The PS lipid vesicle pellet was resuspended in a 0.3 M sucrose solution.

### 4.4. Lipid Vesicle Fusion Mediated by ANXA7

The PS lipid vesicle fusion assay was performed as previously described [82]. Lipid vesicles were first diluted to an *A*540 of 0.6 in a fusion reaction buffer (0.3 M sucrose, 40 mM histidine (pH 6.1), 0.5 mM MgCl2, and 0.1 mM EGTA). Wt or mutant ANXA7 (0.5 mg) was incubated with 0.5 mL of lipid vesicle suspension in a final volume of 1 mL of fusion reaction buffer. Fusion was initiated through the addition of 1 mM Ca^2+^ final and then measured by the change in the turbidity at an absorbance of 540 nm (*A*540) using a recording Hewlett–Packard spectrophotometer for 20 min at room temperature. Fusion was initiated through the addition of the final Ca^2+^ concentrations indicated (0.01, 0.05, 0.4, and 1 mM) and then monitored spectrophotometrically for 20 min. Free Ca^2+^ concentrations were determined as described elsewhere [83] and verified using a Ca^2+^-selective electrode. *p*-values for liposome aggregation assay were determined using a two-tailed Student’s *t*-test.

### 4.5. Protein Modeling

Using the DeepView/Swiss-PdbViewer, v3.7 (Nicolas Guex, Manuel Peitsch, Torsten Schwede, and Alexandre Diemand) software developed at the GlaxoSmithKline and available through public access at Expasy (http://www.expasy.org/spdbv accessed on 1 October 2020), we simulated ANXA7 protein using ANXA1 crystallographic modeling (PDB: 1mcx; Deposition: Luecke H, Rosengarth A). Based on the amino acid composition, the ANXA1 helix-type structure homologous to ANXA7 was assigned to ANXA7 (UniProtKB/SwissProt Entry: P20073), and the C-terminal ANXA7 fragment (184–487) was correctly “threaded” on a structural template of ANXA1 with surrounding Ca ions. Surface (including internal cavities) and electrostatic potential were computed using correspondent options in DeepView/Swiss-PdbViewer software. Additional ANXA7 protein analysis was performed using databases such as Ensembl and Expasy Tools, as well as the Network Protein Sequence Analysis, France [84]. A similar approach was used for the protein simulation of ANXA11 (UniProtKB/SwissProt Entry: P50995) based on the same ANXA1 structural template.

### 4.6. Bioinformatics and Statistics

We used the dominant-negative sequence of ANXA7 through machine learning to predict the amino acids associated with dominant disorders as described by Quinodoz et al. [85].

### 4.7. Cell Culturing and Treatment

Adenoviral vector-based plasmids were constructed with the AdEasy System (a gift from Dr. Bert Vogelstein, Johns Hopkins Oncology Center, Baltimore, MD, USA). Benign prostate epithelial cells (PrECs) and prostatic cancer cells (ATCC, Manassas, VA, USA) were cultured (18 h) after transfection with AdEasy-based adenoviral vectors containing wt-/DN-*ANXA7* or p53 constructs. Briefly, PrEC was grown using a PrEGM Bullet Kit (Cambrex, East Rutherford, NJ, USA) and androgen-sensitive prostate cancer cells; DU145 constructs were grown in RPMI-1640 (Invitrogen—Gibco, Carlsbad, CA, USA) media containing 10% fetal bovine serum along with penicillin–streptomycin. Subconfluent cells were transfected the next day after seeding with a panel (×4) of constructs containing the “empty” vector alone, full-length wt-*ANXA7*, mutant *ANXA7J*, or full-length wt-p53 independently expressing a green fluorescent protein (GFP) expression marker. Infection efficiency was monitored through GFP expression, and optimized amounts of virus were used in order to reach similar levels (70–80%) of efficacy for the different constructs. The dominant-negative mutant construct (DN-*ANXA7J*) contained triple mutations, which were intended to affect the C-terminal residues in the annexin repeats 2, 3, and 4 (E277→Q277, D360-E361→N360-Q361, and D435-D436→N435-N436, respectively). The cells transfected with the vector only were used as a control if not otherwise specified.

### 4.8. Programmed Cell Death (PCD) and Cell Cycling

We performed cell-cycle and apoptosis analyses in parental and transfected (vector alone, WT/DN-*ANXA7J*, or p53) DU145 cells (18 h). Briefly, programmed cell death (PCD) detection and cell-cycle analysis were performed after the transfection of PrEC and DU145 cells with wt-*ANXA7*, DN-*ANXA7J*, and p53 constructs. At 18 h after the infection, both attached and floating cells (1 × 10^6^) were harvested, fixed with 1% paraformaldehyde for 1 h, washed twice with PBS + 0.3% BSA, resuspended with 70% ethanol, and stored at −20 °C overnight. PCD was detected by using an Annexin V-PE Apoptosis Detection Kit I and APO-BRDU Kit (both from BD Pharmingen, San Jose, CA, USA). Both apoptosis detection assays were performed using single GFP-positive cells. Using the Annexin V-PE assay, the cells in the early (phosphatidylserine exposure) and late (membrane permeabilization) stages of PCD were analyzed via flow cytometry (EPICs XL-MCL, Beckman Coulter, Fullerton, CA, USA). Using the APO-BRDU-Kit-based TUNEL (terminal deoxynucleotidedyltransferase dUTP nick end labeling) method, DNA fragmentation in the end stage of PCD was detected via flow cytometry (LSRII, BD Biosciences, San Jose, CA, USA). Cell-cycle analysis was based on propidium iodide (PI) staining after dsRNA removal using DNase-free RNase (Sigma-Aldrich, St. Louis, MO, USA) in the cells fixed in 70% ethanol. ModFit LT (Verity Software House, Topsham, ME, USA) was used for immediate analysis using flow cytometry (EPICs XL-MCL, Beckman Coulter, Fullerton, CA, USA). Experiments were carried out in duplicates or triplicates. Statistical analysis was performed using Student’s *t*-test for independent samples or a two-tailed Z-test for proportions; *p*-values < 0.05 (two-sided test) were considered statistically significant.

### 4.9. RNA Extraction and PCR

Confluent parental and vector-alone-, WT/DN-*ANXA7J*-, and p53-transfected DU145 cells were harvested, and total RNA was isolated with RNAqueous-4 PCR Kit (Ambion, Austin, TX, USA) and used for reverse transcription (SuperScript First-Strand Synthesis System for RT-PCR, Invitrogen, Carlsbad, CA, USA). Multiplex PCR was performed in a 50 µL reaction mixture containing 1 µL reverse transcription product, AmpliTaq Gold with GeneAmp Buffer, and dNTP Mix (Applied Biosystems, Foster City, CA, USA). The following human IP3 receptor primers were prepared and tested in these experiments:

Type1 = forward primer: 5′-CACCGCGGCAGAGATTGACAC-3′; reverse primer: 5′-CCAGCTGCCCGGAGATTTC-3′

Type2 = forward primer: 5′-CTGGGGCCAACGCTAATACTACTT-3′; reverse primer: 5′-GAACCCCGTGATTACCTGTGACTG-3′;

Type3 = forward primer: 5′-GCGGGCCTGTGACACTCTACTTAT-3′; reverse primer: 5′-CGCCGCTCACCAGGGACAT-3′.

For the housekeeping gene beta-actin (as an internal control in multiplex PCR), we used the following primer sequences: forward—5′-CTGGCCGGGACCTGACTGACTACCTC-3′ and reverse—5′-AAACAAATAAAGCCATGCCAATCTCA-3′.

PCR procedure and cycling conditions (annealing temperature of 58–60 °C, ×30–38 cycles on GeneAmp PCR System 9600, Perkin Elmer, Wellesley, MA, USA) were standardized for the RNA amount and the linear range of amplification. PCR protocol comprised an initial denaturation step at 94 °C for 60 s and a final extension step at 72 °C for 600 s. PCR products (10 µL) were mixed with 5× DNA Gel Loading Solution (Quality Biological, Gaithersburg, MD, USA), electrophoresed at 200 V in polyacrylamide 4–20% TBE gels (Invitrogen, Carlsbad, CA, USA), and visualized using ethidium bromide staining.

### 4.10. cDNA Microarray Experiment and Data Mining Using Gene Pattern and Ingenuity Pathway Analysis

*Preparation and labeling of RNA:* Total RNAs from prostate normal and cancer cells, either parental or transfected with either vector alone or wt-*ANXA7*, *ANXA7J,* or p53 as a positive control were prepared following the method of [86] and were subjected to DNAse 1 digestion to eliminate genomic DNA contamination. Two rounds of purification of poly A+ RNA from total RNA were performed using the mRNA isolation kit from Invitrogen, as recommended by the manufacturer. The quality of the RNA was tested by running a formaldehyde-denatured agarose gel and quantitated by measuring the optical density at 260 nm. A ^32^P-labeled cDNA probe was synthesized from 1 μg of poly A+ RNA from control and tumor samples using MMLV reverse transcriptase, dNTP mix, and CDS primer mix comprising the oligonucleotide sequences for the 1200 cancer-related genes spotted on the Atlas cDNA microarray. The reaction was carried out in a thermocycler set at 50 °C for 25 min and terminated through the addition of 0.1 M EDTA, pH 8.0, and 1 mg/mL glycogen. The ^32^P-labeled cDNA probe was then purified from unincorporated ^32^P-labeled nucleotides by using a CHROMA SPIN-200 column (Takara Bio, Ann Arbor, MI, USA), as recommended by the manufacturer. The human Atlas cDNA expression array containing 1200 cancer-related genes on a nylon membrane was prehybridized using Express Hyb (Takara Bio, USA) at 68 °C for 1 hr and hybridized overnight at 68 °C with the denatured and neutralized ^32^P-labeled cDNA probe. The membrane was washed three times with 2 × SSC, 1% SDS at 68 °C for 30 min each and twice with 0.1% SSC, 0.5% SDS at 68 °C for 30 min each. The Atlas array was exposed overnight, and the results were compared with the known distribution of genes.

*Imaging and quantitation of the cDNA microarray:* Imaging data from the Storm PhosphorImager were downloaded into a Microsoft Excel spreadsheet. The ratios of duplicate data points to the ubiquitin standard were determined. Data were then analyzed using the Stanford University ScanAlyze software (Version 2006). These data were also evaluated in parallel with the PSCAN program for point identification and with the JMP program for graphical organization.

*Statistical Data mining from cDNA arrays:* Arrays representing DU145 cells transfected with the wt-*ANXA7*, DN-*ANXA7J*, or p53 were cross-compared after data normalization and filtration using more advanced options of corresponding software. Normalization was determined by assuming the equivalent global hybridization of all test and reference probes. The adjusted intensities were calculated as spot intensities minus background values for spots, multiplied by normalization coefficients. Data were filtered to remove values from poorly hybridized cDNAs with low-intensity levels based on the two thresholds: ratio (2-fold change) and the actual difference between adjusted intensities depending on the background.

## 5. Conclusions

In conclusion, the regulation of IP3Rs by ANXA7 appears to be a general mechanism to potentiate calcium release in growing cells. mTOR activity is, by itself, very complex and relies on multiple signals from mitogenic growth factors, nutrients, cellular energy levels, and stress conditions. Therefore, IP3R activity may be modulated by all these different signals. In the present study, we identified the amino acid residues that play an important role in regulating IP3 receptor expression and cell survival signaling by performing site-directed mutagenesis studies on human ANXA7. The results suggest that glutamic and aspartic acid are involved in calcium and phospholipid binding as determinants of ANXA7 action on IP3 receptor expression and prostate cancer cell survival acting via the PI3K/mTOR pathway. Our study lends further support to the emerging concept of crosstalk between Ca2+ signaling and proliferation pathways and thus provides another way through which Ca2+ signals are finely encoded within cells.

We discovered this gene (*ANXA7*) during the early 1990s and have been working to understand its cellular functions and functional role in different cancers. However, in this study, we first uncovered the structure–function relationship within the primary sequence of ANXA7 and how the sequential mutations alter its basic Ca signaling and other cellular functions acting via the PI3K/mTOR pathway. Although we tried to logically and sequentially address the functional role of ANXA7 in regulating cell–cell membrane fusion through charge distribution, this study has certain limitations. In this study, we focused on the regulation of IP3Rs by ANXA7 and did not investigate other potential mechanisms of calcium signaling regulation or the role of other annexin family members in tumor suppression. Furthermore, the study was conducted in vitro, and the results may not necessarily translate to the complex environment of living organisms. More research is needed to validate our findings in animal models or human subjects.

Exemplifying the member role of the annexin family, ANXA7 could translate conventional annexin properties such as PS/Ca binding and exocytosis into tumor suppression mechanisms. A protein known to regulate how cells process calcium also appears to be a tumor suppressor, adding to the potential that therapies directed at cellular metabolism may help suppress tumor growth. As a valuable therapeutic target in cancer and other pathologies, ANXA7 appears to be a rapamycin-like endogenous regulator that can balance cell growth and survival in homeostasis and tumorigenesis instead of delivering a potentially detrimental “chemical knockout” of the mTOR/Akt/PI3K pathway components. ANXA7-associated apoptotic and pharmacological manipulation has the potential to meet the urgent need for drugs that will preferentially kill cancer cells and shield normal cells, thus presenting a new mode of action that is vital in the management of cancer therapy.

## Figures and Tables

**Figure 1 ijms-24-08818-f001:**
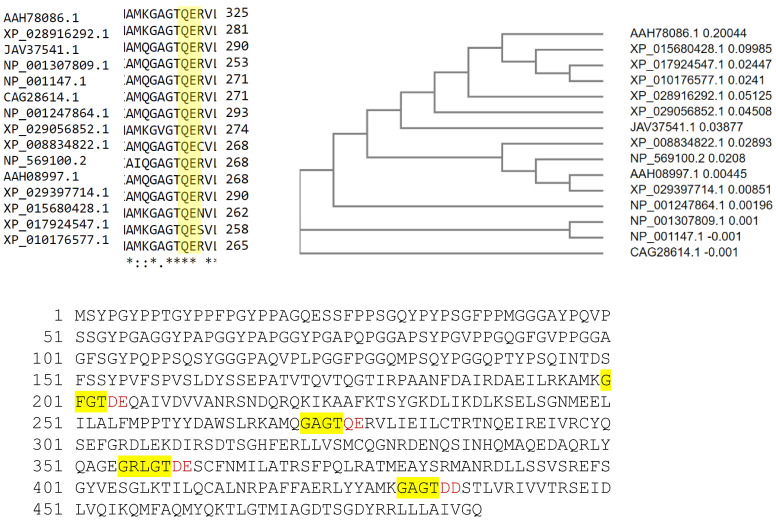
Amino acid sequence alignment of all annexins in many species. The sequence alignment shows that the endonexin-fold motif (GX(X)GT) in each repeat in the C-terminal domain is highly conserved among all annexins in many species. The amino acids that were modified in each repeat via site-directed mutagenesis are shown in red. * Indicates conserved amino acids in the endonexin fold.

**Figure 2 ijms-24-08818-f002:**
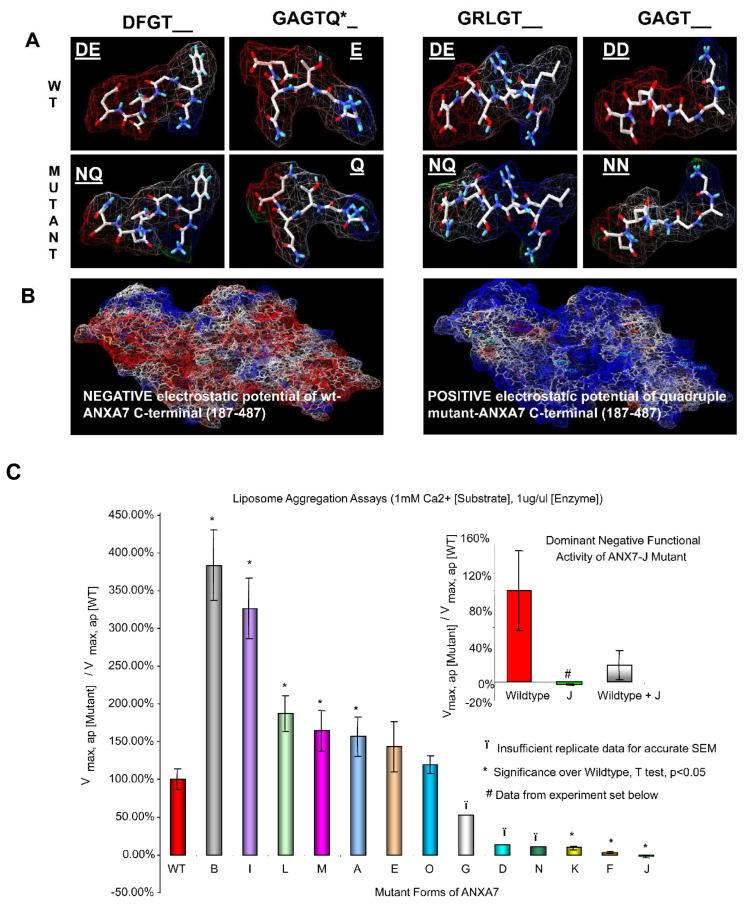
Simulation of ANXA7 structural model with the electrostatic potential distribution. Surface (including internal cavities and N-terminal) and electrostatic potential were computed using correspondent options in the DeepView/Swiss-PdbViewer. Electrostatic potential (negative: −1.800, dark red and positive: +1.800, dark blue) on the external surface for both N-terminal (1–20) and C-terminal (187–487) with internal cavities was computed using the Coulomb method with partial atomic charges for dielectric constant (solvent) and is shown with dotted lines (both (**A**,**B**)) or filled triangles ((**B**), N-terminal and internal cavities only). The bipolar charge distribution is shown in the N-terminal fragment alone (Panel (**A**)) and in the N-terminal enclosed into the C-terminal core in both wt-ANXA7 and DN-ANXA7J models (Panel (**B**)). The C-terminal model for the wt-ANXA7 (Panel (**B**)) is viewed from the bottom and for the mutant DN-ANXA7J from the top. Selected residues are displayed with sidechains and colored based on their putative affinity to Ca or PS (as indicated in Figure 1 legend). N-terminal residues as well as some additional C-terminal residues at the Ca- and PS-binding sites are colored using CPK. Surrounding Ca ions are shown as green balls. (**C**) PS liposome aggregation profiles of the wt-ANXA7 versus multiple mutant ANXA7 constructs. The pTRC vector-based plasmids for wt-ANXA7 and mutant ANXA7 were constructed using the AdEasy System. Mutations affected polar C-terminal residues in consequential repeats of the wt or normal status, and the combinatorial DN-ANXA7J contained triple mutations. PS liposome aggregation was assessed using a spectrophotometry-based assay and is presented as fusogenic rate (or velocity) at the fixed 1 mM Ca concentration relative to the wt-ANXA7 percentage levels: Vmax, ap (mutant)/Vmax, ap (WT); (*) when *p* < 0.05.

**Figure 3 ijms-24-08818-f003:**
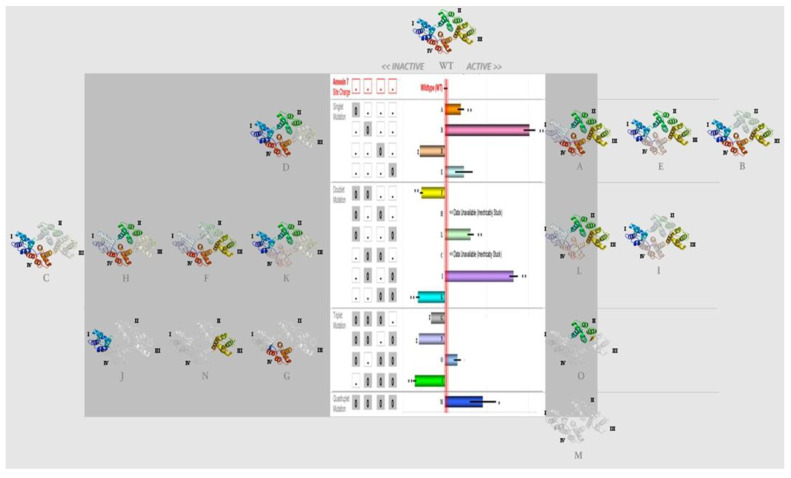
Membrane fusion functionality of ANXA7 Ca2+-binding mutants. Composite bar chart of all mutant ANXA7 constructs (at 1.0 mM Ca2+ concentration) grouped by tuples: Singlet mutations **A**, **B**, and **E** were active, except mutant **D** (3); doublet mutations **F**, **K**, **L**, and **I** were either active or inactive, except inconclusive mutant **C** (2 and 3) and mutant **H** (1 and 3); triplet mutations **G**, **N**, and **J** were inactive, except mutant **O** (1, 3, and 4). INXS: mutant **C** (2 and 3) and mutant **H** (1 and 3) were inextricably stuck to the *E. coli* membrane and 90% acetone delipidation treatment denatured protein functionality. Roman letters (I–IV) indicate the four repeats of the C-terminal region of the ANXA7 protein. The colorless repeats are the mutated regions.

**Figure 4 ijms-24-08818-f004:**
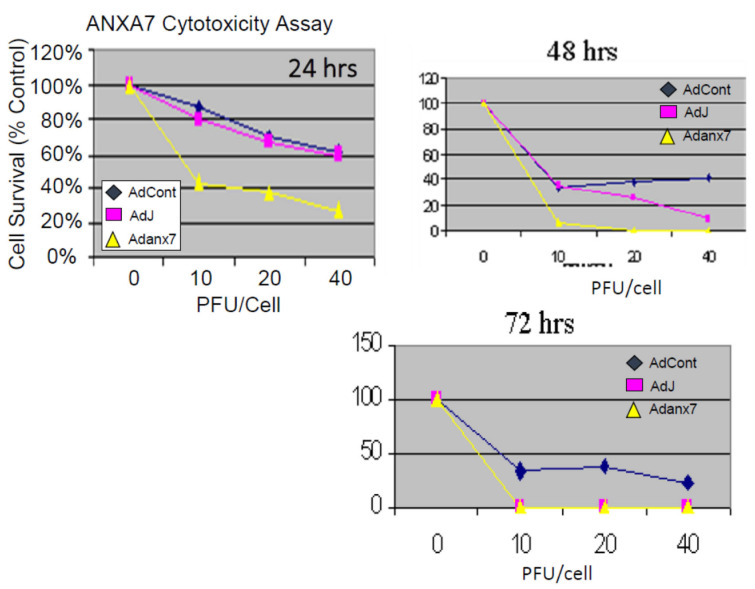
Dose dependence of ANXA7 cytotoxicity. Dose-dependent (infection of viruses per cells; PFU/cells 0, 10 20, and 40) relationship between ANXA7 concentration and cell numbers were assessed using trypan blue dye exclusion assay. DU145 cells overexpressing wt-*ANXA7*, DN-*ANXA7J*, and Ad-CMV (control) were grown for 24, 48, and 72 h and stained with trypan blue, and the viable cells that excluded the dye were counted.

**Figure 5 ijms-24-08818-f005:**
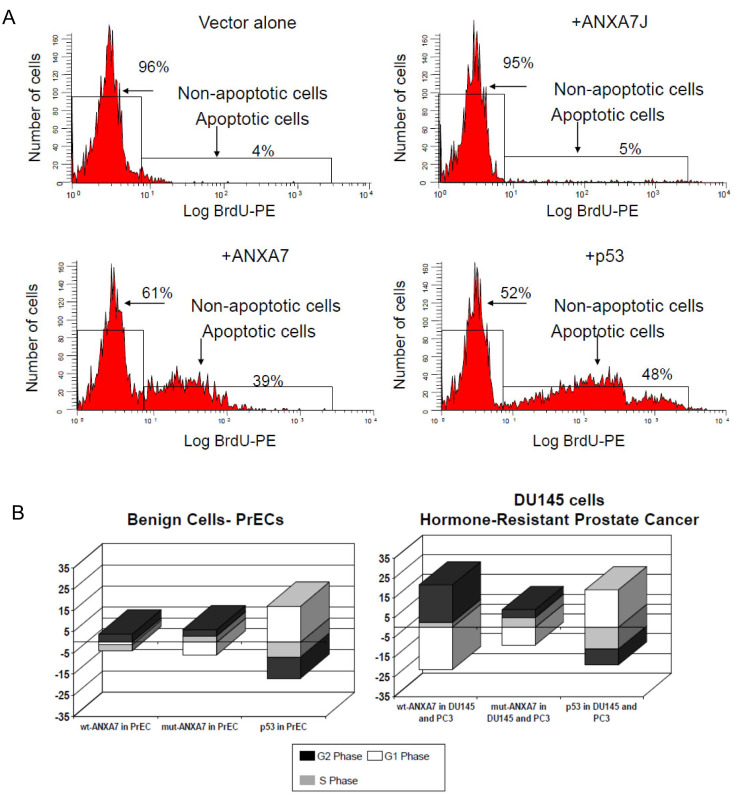
Effect of ANXA7 on cell cycle in DU145 cells: (**A**) Flow cytometric analysis of DU145 prostate cancer cells transfected with adenovirus vector alone or wild-type *ANXA7* or dominant-negative *ANXA7J.* (**B**) Relative cell numbers in different phases compared with vector in each category (delta %).

**Figure 6 ijms-24-08818-f006:**
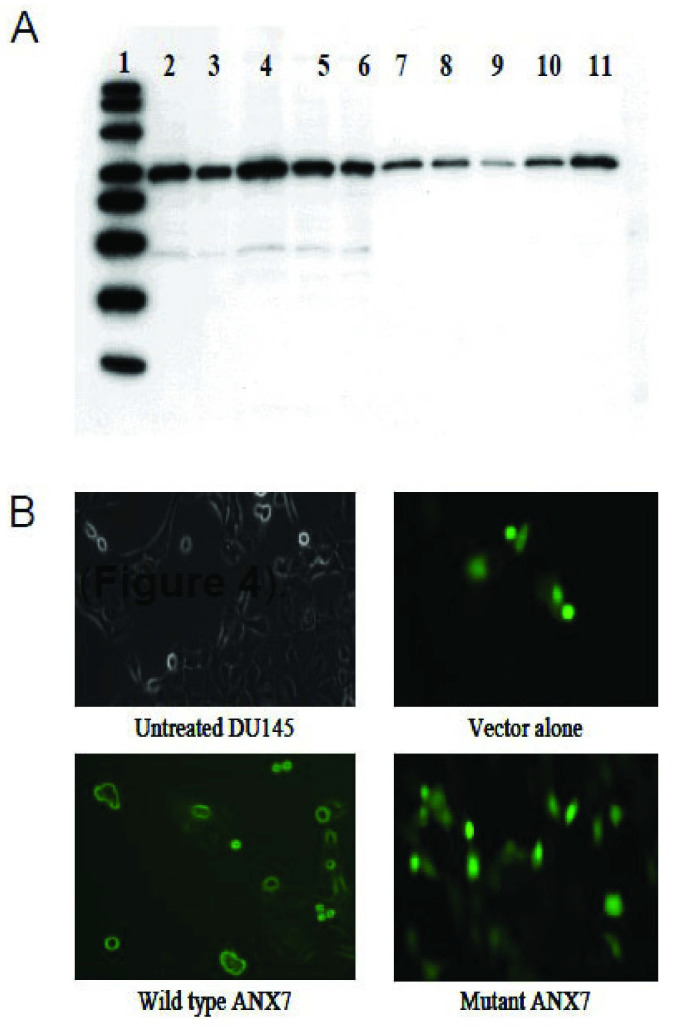
Effect of ANXA7 on cytochrome c release and DU145 cell morphology: (**A**) Western blot of cytochrome c release from DU145 prostate cancer cell line transfected with adenovirus vectors. Lanes: 1, MagicMark Standard; 2–6, cytosol fractions of parental control, Ad control, AdAnxA7, AdJ, and p53, respectively; 7–11, mitochondrial fractions of parental control, Ad control, AdAnxA7, AdJ, and p53, respectively. (**B**) Fluorescent photomicrographs of DU145 cells (20× magnification) overexpressing Ad-wt-*ANXA7*, Ad-DN-*ANXA7J*, and Ad-CMV (control) grown for 24 h at 10 pfu/mL The green fluorescence is due to the expression of the green fluorescent protein.

**Figure 7 ijms-24-08818-f007:**
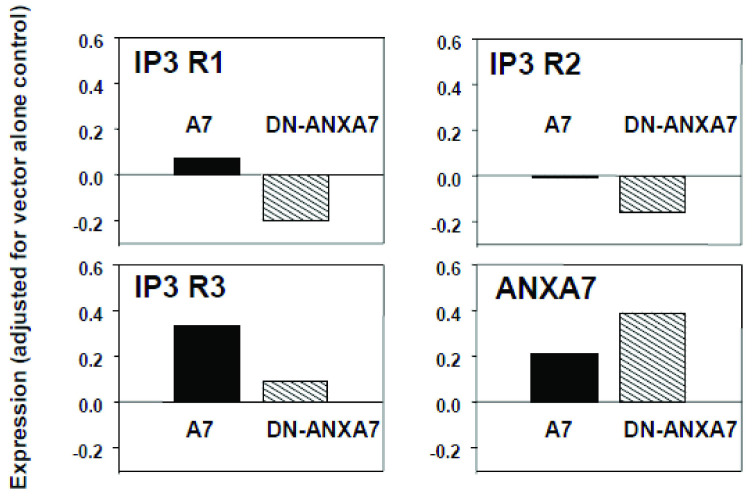
Influence of wild-type *ANXA7* and dominant-negative *ANXA7J* mutant on IP3 receptor subtype expression. Measurement of mRNA levels of *ANXA7* and IP3 receptor (types 1, 2, and 3) in tumor cells treated with adenovirus vector alone, wild-type, and dominant-negative mutant *ANXA7J* using quantitative RT-PCR.

**Figure 8 ijms-24-08818-f008:**
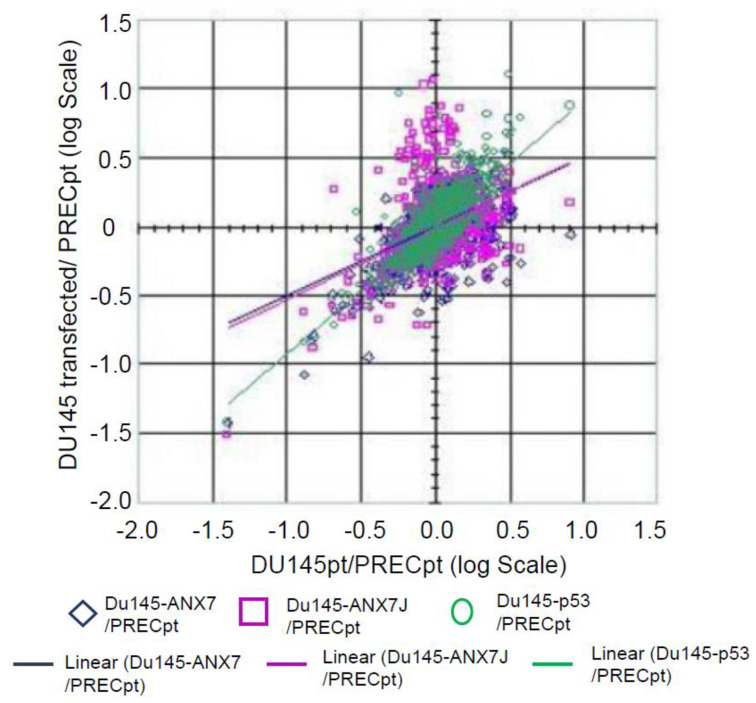
Gene expression profiling of DU145 cells transfected with DN-*ANXA7J*, wt-*ANXA7*, or P53. The expression levels in DU145 cells (as log ratio to expression in normal PREC prostate cell line) after treatment with DN-*ANXA7J*, wt-*ANXA7*, P53, or vector alone are shown. The x-axis shows the average expression level of the respective genes in the metastatic DU145 and PREC cells. The transfected DU145 cells are depicted with magenta diamonds (for DN-*ANXA7J*), blue diamonds (for wt-*ANXA7*), and green diamonds (for P53). The improvement in the expression levels is reflected in the movement of the expression in the transfected cells towards the *x*-axis.

**Figure 9 ijms-24-08818-f009:**
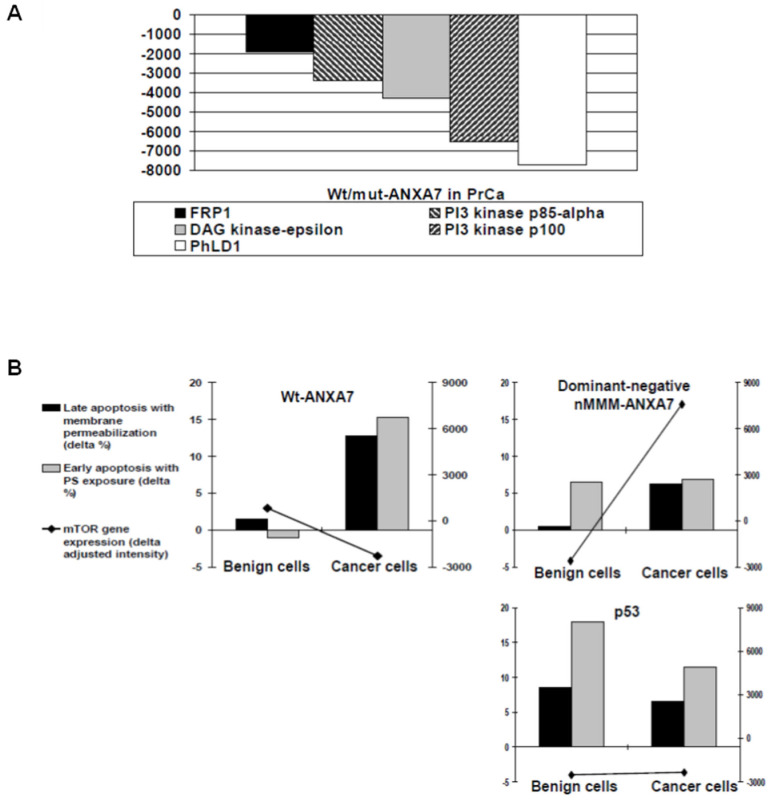
(**A**) Comparative lipid-relevant gene expression profiles with the introduction of wt-*ANXA7* and DN-*ANXA7J* in averaged PrCa array. cDNA microarray analysis was performed using Atlas Human Cancer 1.2 arrays and corresponding software AtlasImage 2.01 (Clontech, Palo Arto, CA, USA). Averaged wt- or mut-*ANXA7* arrays for PrCa were created using prostate cell lines (LNCaP, DU145, and PC3). Wt/DN-*ANXA7J* ratio was assessed using the actual difference between the adjusted intensities after subtraction of the external background and the global normalization based on the sum method. Each of the presented genes was found on the outliers lists either in PrCa with the following criteria: R > 2 and difference threshold >4000. (**B**) Apoptotic rates including PS exposure with corresponding *mTOR* gene expression in response to wt/DN-*ANXA7J* or p53 in benign versus cancerous and prostate cells. Type I PCD rates as early apoptosis with PS exposure (grey columns) and late apoptosis with membrane permeabilization (black columns) by ANXAV-PE were compared with the vector in each category and presented as delta % (left scale). mTOR gene expression was compared with the averaged CELL array and presented as the actual difference between the adjusted intensities after subtraction of the external background and the global normalization based on the sum method (black line, right scale).

**Table 1 ijms-24-08818-t001:** Combinatorial mutations.

Singlet (Mutation Site)	Doublet (Mutation Sites)	Triplet (Mutation Sites)	Quadruplet (Mutation Sites)
A (1)	C (2, 3)	G (1, 2, 3)	M (1, 2, 3, 4)
B (2)	F (1, 2)	J (2, 3, 4)	
D (3)	H (1, 3)	N (1, 2, 4)	
E (4)	I (2, 4)	O (1, 3, 4)	
	K (3, 4)		
	L (1, 4)		

**Table 2 ijms-24-08818-t002:** List of the genes that are most affected by transfection of *ANXA7*, *ANX7AJ*, or P53 in terms of reverting back to expression levels in normal prostate cells.

	Corrective Effect of:		
Gene Name	*ANXA7*	*ANX7AJ*	P53
ribosomal protein S6 kinase II alpha 3 (S6KII-alpha 3); ribosomal S6 kinase 2 (RSK2); insulin-stimulated protein kinase 1 (ISPK1)	3.5	1.6	−0.1
bcl2 homologous antagonist/killer (BAK)	−0.2	0.2	3.3
MHC class II HLA-DR-beta (DR2-DQW1/DR4 DQW3) precursor	0.1	0.1	3.1
B4-2 protein	2.8	1.5	0.5
integrin alpha E precursor (ITGAE); mucosal lymphocyte-1 antigen; hml-1 antigen; CD103 antigen	2.7	0.0	−0.1
integrin alpha 8 (ITGA8)	2.5	0.7	−0.1
DNA fragmentation factor 45 (DFF45)	2.5	0.4	0.9
hyaluronan receptor (RHAMM)	2.5	0.0	−0.2
CDC25C; M-phase inducer phosphatase 3	2.4	−0.1	−0.2
cadherin 6 precursor (CDH6); kidney cadherin (K-cadherin)	2.4	0.4	0.1
cation-independent mannose-6-phosphate receptor precursor (CI man-6-P receptor; CI-MPR); insulin-like growth factor II receptor (IGFR II)	0.9	−0.1	2.4
clone PO2ST9 (brain striatum)	−0.2	−0.3	2.3
sonic hedgehog (SHH)	2.1	0.0	0.1
transforming growth factor beta2 precursor (TGF-beta2; TGFB2); glioblastoma-derived T-cell suppressor factor (G-TSF); bsc-1 cell growth inhibitor; polyergin; cetermin	2.1	0.3	0.2
c-myc oncogene	0.0	0.0	2.0
HLA-DR antigen-associated invariant subunit	0.1	2.0	−0.2
heparin-binding growth factor 2 precursor (HBGF2); prostatropin; basic fibroblast growth factor (BFGF; FGFB; FGF2)	0.9	−0.7	2.0
ras-like protein TC10	2.0	0.5	−0.1
interferon-induced guanylate-binding protein 1; guanine nucleotide-binding protein 1	−0.5	1.9	−0.1
skeletal muscle phosphorylase B kinase gamma catalytic subunit	1.9	−0.1	−0.2
interleukin-7 (IL-7)	0.1	1.9	−0.2
cyclin-dependent kinase regulatory subunit (CKS2)	1.9	0.5	1.1
eukaryotic translation initiation factor 3 beta subunit (EIF-3 beta); EIF3 P116	−0.1	−0.2	1.9
semaphorin; CD100	1.9	0.3	−0.3
ephrin type-B receptor 4 precursor; tyrosine-protein kinase receptor HTK	0.9	1.9	0.3

## Data Availability

All data described in this manuscript are entirely available in this article. We also provided all websites or web portals where additional data are available.

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
