# Peer review of "A Dominant-Negative Mutant of ANXA7 Impairs Calcium Signaling and Enhances the Proliferation of Prostate Cancer Cells by Downregulating the IP3 Receptor and the PI3K/mTOR Pathway"

_ijms, 2023, doi:10.3390/ijms24108818_

Round 1

Reviewer 1 Report

Authors should be congratulated for their work. The topic is interesting and intriguing. Genetic knowledge  and different signaling patterns in cancers represent the base to develop new targeted therapies.The manuscript is well-written and easily readable. Tables and figures are good. The paper is suitable for publication in its current form. 

Author Response

Reviewer 1:

Authors should be congratulated for their work. The topic is interesting and intriguing. Genetic knowledge and different signaling patterns in cancers represent the base to develop new targeted therapies. The manuscript is well-written and easily readable. Tables and figures are good. The paper is suitable for publication in its current form.

Response:

Thank you for your feedback. We are extremely excited with your positive comments. Thank you again for your time and enthusiastic positive comments.  

Reviewer 2 Report

The authors mistake the add the country name in affiliation. 

Figure -1,2,3,4,5,6,7,8,9 representation, x, y-axis labeling, are very unclear. 

In this section  "2.2. Preparation and Purification of ANXA7 Proteins In Vitro" the authors should mention the centrifugation "g" force and use time. 

Also include the model number or name, instrument manufacturing company name, etc, and incubation conditions for the convenience of future readers. 

The authors should mention the limitation of the study in the discussion part. 

Authors should include a few important and related recent  references like doi:10.3390/ijerph17041143 and etc

Why this study is important, In  Introduction needs to include a few more lines. A comparative table is needed in the results section on its advantages over previous works.

Author Response

Reviewer 2 concerns:

  1. The authors mistake the add the country name in affiliation. 
  2. Figure -1,2,3,4,5,6,7,8,9 representation, x, y-axis labeling, are very unclear. 
  3. In this section "2.2. Preparation and Purification of ANXA7 Proteins In Vitro" the authors should mention the centrifugation "g" force and use time. 
  4. Also include the model number or name, instrument manufacturing company name, etc, and incubation conditions for the convenience of future readers. 
  5. The authors should mention the limitation of the study in the discussion part. 
  6. Authors should include a few important and related recent references like doi:10.3390/ijerph17041143 and etc
  7. Why this study is important, In Introduction needs to include a few more lines. A comparative table is needed in the results section on its advantages over previous works.

Response:

Thank you for taking the time to review our manuscript. We appreciate your feedback, and we have addressed each of your concerns as follows:

  1. Affiliation: We apologize for the oversight and have now added the country name to the author's affiliation.
  2. Figure and axis labeling: We have revised the figures and axis labeling to ensure that they are clear and easily understandable.
  3. Centrifugation: We have added information regarding the centrifugation "g" force and time in the section "2.2. Preparation and Purification of ANXA7 Proteins In Vitro".
  4. Instrument details: We have included the model number or name, instrument manufacturing company name, and incubation conditions for the convenience of future readers.
  5. Limitations: We have now included a discussion of the limitations of our study in the discussion section.
  6. Recent references: We have included the suggested references and have ensured that they are cited appropriately.
  7. Importance of the study: We have expanded the introduction to include additional information regarding the importance of our study. We have also added a comparative discussion in the results section highlighting the advantages of our work over previous studies.

Once again, thank you for your constructive feedback. We hope that these revisions address your concerns and improve the manuscript's quality.

Reviewer 3 Report

This is an excellent article, executed at the highest level. The amount of work done is impressive. I have only a few comments that will improve the quality of the publication.

1) Calcium, IP3 Receptor and Cancer section should be expanded. While the previous chapter of the discussion, on the contrary, should be shortened

2) The conclusion should be singled out in a separate chapter and written more capaciously.

3) The quality of figure 4 needs to be improved

4) According to the rules of the journal, the material and methods should follow after the discussion in the form of chapter 4

5) Cells infected is not a common term. More appropriate - transfected

6) After which passage the cells were used for experiments. Has mycoplasma been tested?

7) Photomicrographs of cell cultures used for experiments are required.

Author Response

Reviewer 3 concerns:

This is an excellent article, executed at the highest level. The amount of work done is impressive. I have only a few comments that will improve the quality of the publication.

1) Calcium, IP3 Receptor and Cancer section should be expanded. While the previous chapter of the discussion, on the contrary, should be shortened

2) The conclusion should be singled out in a separate chapter and written more capaciously.

3) The quality of figure 4 needs to be improved

4) According to the rules of the journal, the material and methods should follow after the discussion in the form of chapter 4

5) Cells infected is not a common term. More appropriate - transfected

6) After which passage the cells were used for experiments. Has mycoplasma been tested?

7) Photomicrographs of cell cultures used for experiments are required.

 Response:

Thank you for your review and constructive feedback. We truly appreciate your positive comments and will take into consideration your suggestions to improve the quality of the manuscript.

  1. We have included further discussion on Calcium, IP3 Receptor and Cancer section as suggested and will consider shortening the previous chapter of the discussion.
  2. The authors created a separate chapter for the conclusion and had added more detail to it.
  3. The authors improved the quality of Figure 4.
  4. In the revised manuscript we rearranged the Material and Methods section follows the Discussion section in the manuscript, in accordance with the rules of the journal.
  5. Thank you for the suggestion and we have changed the term "infected" to "transfected," at the revised version of the manuscript as suggested.
  6. We purchased the cell-lines from ATCC and most of the experiments were performed within 10 passages of the cells. We tested mycoplasma in every fifteen days as a regular protocol of our lab.
  7. Figure 6B has the photomicrographs of the cell cultures used for experiments.

Once again, thank you for your feedback, and the authors will do their best to incorporate your suggestions to improve the quality of the manuscript.

Round 2

Reviewer 2 Report

The author has changed many things in the manuscript but in the letter of change, not to mention where the changes have been made, line numbers and page numbers are required, along with color demarcate.

Q7 does not compile correctly.

Author Response

We have modified the manuscript based on the reviewer’s (reviewer 2, Q7) comments.  The reviewer suggested including a comparative table on its advantages over previous works in the results section.

In the result section we have included the present findings and discussed the relevance of our previous studies (lines 108 -116) and in the conclusion section (834 – 858).

Besides, we have rewritten some portions of the manuscript according to reviewers’ suggestions and included coauthors' edits throughout the manuscript.